# A Hybrid Deep Learning Model with Data Augmentation to Improve Tumor Classification Using MRI Images

**DOI:** 10.3390/diagnostics14232710

**Published:** 2024-11-30

**Authors:** Eman M. G. Younis, Mahmoud N. Mahmoud, Abdullah M. Albarrak, Ibrahim A. Ibrahim

**Affiliations:** 1Faculty of Computers and Information, Minia University, Minia 61519, Egypt; eman.younas@mu.edu.eg (E.M.G.Y.); mahmodnaser92pg@fci.s-mu.edu.eg (M.N.M.); 2College of Computer and Information Sciences, Imam Mohammad Ibn Saud Islamic University (IMSIU), Riyadh 13318, Saudi Arabia; amsbarrak@imamu.edu.sa

**Keywords:** brain tumor, CNN, EfficientNetV2B3, KNN

## Abstract

Background: Cancer ranks second among the causes of mortality worldwide, following cardiovascular diseases. Brain cancer, in particular, has the lowest survival rate of any form of cancer. Brain tumors vary in their morphology, texture, and location, which determine their classification. The accurate diagnosis of tumors enables physicians to select the optimal treatment strategies and potentially prolong patients’ lives. Researchers who have implemented deep learning models for the diagnosis of diseases in recent years have largely focused on deep neural network optimization to enhance their performance. This involves implementing models with the best performance and incorporating various network architectures by configuring their hyperparameters. Methods: This paper presents a novel hybrid approach for improved brain tumor classification by combining CNNs and EfficientNetV2B3 for feature extraction, followed by K-nearest neighbors (KNN) for classification, which has been described as one of the simplest machine learning algorithms based on supervised learning techniques. The KNN algorithm assumes similarities between new cases and available cases and assigns new cases to the category that most closely resembles the available categories. Results: To evaluate the recommended method’s efficacy, two widely known benchmark MRI datasets were utilized in the experiments. The initial dataset consisted of 3064 MRI images depicting meningiomas, gliomas, and pituitary tumors. Images from two classes, consisting of healthy brains and brain tumors, were included in the second dataset, which was obtained from Kaggle. Conclusions: In order to enhance the performance even further, this study concatenated the CNN and EfficientNetV2B3’s flattened outputs before feeding them into the KNN classifier. The proposed framework was run on these two different datasets and demonstrated outstanding performance, with accuracy of 99.51% and 99.8%, respectively.

## 1. Introduction

Among all causes of death, cancer ranks sixth in the world. It is an essential pathological condition. Brain tumors are thought to be among the deadliest malignancies, with poor survival rates [1]. The tumor shape, texture, location, and other factors can significantly affect the types of tumors that occur in the brain, which include gliomas, pituitary tumors, and meningiomas [2]. The rates of occurrence for tumors of the brain during clinical monitoring are approximately 45 percent, 15 percent, and 15 percent, respectively. Doctors can obtain diagnoses and predict the survival of patients based on the type of tumor [3]. Additionally, they make decisions by following the appropriate care before surgery. Radiotherapy and chemotherapy adopt a ’wait and see’ strategy, avoiding invasive procedures. Moreover, a crucial part of treatment planning is tumor grading [4].

Brain tumors can be detected in 2D and 3D using magnetic resonance imaging (MRI), a nonsurgical, quick, and easy medical imaging technique. It is one of the most widely accepted techniques for the identification and detection of cancer, focusing on ultra-high-definition images of brain tissue [5].

However, identifying the type of cancer using tomography images may be a challenging, inaccurate, and time-consuming exercise, requiring highly specialized physician experience and entailing a laborious process [6]. Tumors can appear in many different shapes, and the images may not contain enough discernible landmarks to aid in medical assessment. The traditional diagnostic method of histopathology involves detecting the tumor type via the microscopic examination of a tumor biopsy on a tumbler slide. Achieving a comprehensive analysis will allow the patient to begin the best treatment immediately, extending their life [7]. Accordingly, this highlights the pressing need for artificial intelligence (AI) to provide and expand a new and modern computer-assisted diagnosis (CAD) system. This system will alleviate the workload during the analysis of tumor types and act as a supporting device for medical doctors and radiologists. The proposed concept in this paper seeks to provide implementation information for solutions for the classification of tumors using classical and hybrid techniques combining convolutional neural networks (CNNs) with classical machine learning. We assessed the proposed technique using an MRI brain tumor dataset including three types of brain tumors (meningiomas, gliomas, and pituitary tumors), as well as images classified as ’tumor’ or ’non-tumor’ [8]. The nominal objective of this research is to explore how artificial intelligence can be used to determine the type of tumor in order to identify the appropriate treatment method, as, in most medical cases, brain surgery can be life-threatening and lead to serious health issues for the patient.

In summary, manual diagnosis is often inaccurate. Furthermore, the treatment of brain tumors frequently presents problems because it can impair a patient’s ability to respond effectively during surgery and lower their chances of survival [9]. However, accurate diagnosis can benefit patients, allowing them to begin the appropriate course of treatment immediately. Consequently, there is a critical need to develop innovative machine diagnosis systems via artificial intelligence (AI). These systems aim to alleviate the burden of patient diagnosis and tumor classification, providing a helpful tool for radiologists and physicians [10]. By decreasing the impacts associated with tumor diagnosis and classification, these systems aim to benefit doctors and radiologists [10].

The most popular systems use convolutional neural networks (CNNs) with several layers based on linear equations between matrices known as convolutions. In addition, these CNNs have fully connected, nonlinear, pooling, and conventional layers. Meanwhile, fully connected, traditional layers consist of parameters and nonlinear and pooling layers [11].

EfficientNetV2 is a new type of CNN designed to improve the speed and effectiveness compared to previous structures. It has provided the strongest results and improved the effectiveness in classifying many types of tumors using ImageNet. In brief, EfficientNetV2B3 may offer a suitable model for medical image classification, requiring the input of 32 × 32 images and 14.5M parameters and resulting in accuracy of 95.8% [12].

EfficientNetV2 surpasses EfficientNet to improve the speed, effectiveness, and productivity. It was created by employing a set of scaling parameters (profundity, determination, and width). EfficientNetV2 is much faster than past and current state-of-the-art models, and it is much smaller (up to 6.8 times) [13]. The input image size and regularization parameters are user-defined, where EfficicentNetV2 employs three types of regularization: dropout, RandAugment, and mistakes [14].

The CNN model, which comprises five blocks of layers with two dropout layers and max pooling after each block, is combined with EfficientNetV2B3 in the proposed framework. It has a primary block of 32 filters and individual blocks of 64, 128, and 256 filters [15].

EfficientNetV2B3 combines dropout, pooling, and batch normalization to form a 24-layer architecture consisting of one fully connected layer, one softmax layer, and 22 convolutional layers. Eight blocks constitute the convolutional layers, and the sizes of the inputs of the images are changed to 32 by 32 pixels [16]. When the output of each model integrates two layers, each having 64 and 16 units, the proposed model performs better [17,18].

Many algorithms are employed in brain tumor classification, such as KNN and deep learning methods. In this research, a hybrid framework is proposed, combining CNN and EfficientNetV2B3 for feature extraction and employing KNN for the classification of images.

The structure of this paper is as follows. Section 2 presents related research on the classification of brain tumors. Section 3 presents the problem definition. Section 4 illustrates the proposed approach, including the dataset, image prepossessing, data augmentation, feature extraction, classification models, implementation, and evaluation metrics. Section 5 presents the results of our experiments. Section 6 presents the ablation study, and a discussion is presented in Section 7. Finally, the conclusions regarding the proposed hybrid method are presented in Section 8.

## 2. Related Work

Image processing techniques play a significant role in medical applications by assisting in the detection of anomalies and diagnosis of diseases. This includes medical images obtained through imaging techniques such as CT scans, MRI, and X-rays. MRI, in particular, has a significant advantage over other diagnosis methods because it does not expose patients to radiation [19].

Across several domains, including computer vision, robotics, and computer-aided diagnosis, deep learning has become a powerful tool. This paves the way for its integration with biomedical image processing, which could result in significant improvements in medical care [20].

Deep learning models acquire the capability to learn several representation and abstraction levels across a large scale because they are fed with raw data. They possess many advantages over existing machine learning techniques, which are restricted to processing only real-life image data, are slower, and require a large amount of effort to adjust the functionalities [21].

CNNs are among these types. They are most commonly utilized for image, video, and speech recognition. The biological functions of the visual system in animals have served as inspiration for CNNs. The CNN’s ability to identify patterns in images has allowed it to be successfully used in image processing [22].

Convolutional layers are composed of extracted features such as colors and edges, and they rely on utilizing a learnable kernel. The cores’ spatial dimensions are small but they are distributed across the input area depth. Layers spread each filter horizontally to create a feature map using the spatial dimensions of the input. The pooling layer is used to decrease the number of variables and the mathematical model’s complexity while lowering the dimensions of the features. Many algorithms have been employed in brain tumor classification, like KNN [23].

Cheng et al. [24] proposed a model consisting of GLCM and BOW, which was applied to a dataset with three different types of brain tumors and obtained precision of 91.28%.

Ismael et al. [25] presented a model based on a neural network algorithm and used statistical features; it obtained 91.9% precision. Its specificity for pituitary tumors was 95.66%, while it was 96% and 96.29% for meningiomas and gliomas, respectively.

Afshar et al. [26] proposed a model for the categorization of brain tumors with rough boundaries, using an additional pipeline as input to improve CapNet’s focus, with accuracy of 90.89%. In [10], Deepak et al. proposed a comprehensive framework based on the pre-trained GoogleNet for feature extraction on an MRI dataset. Gull et al. [27] proposed a model that utilized AlexNet and VGG-19 for brain tumor classification and achieved accuracy of 98.50% and 97.25%, respectively, for VGG-19 and AlexNet. Mondal et al. [28] compared the results of various CNN models (such as DenseNet201 and ResNet50) and achieved accuracy of 98.33%.

## 3. Problem Definition

Determining a tumor’s grade and type is essential, particularly at the beginning of the treatment plan. The accurate detection of abnormal tissue is essential for diagnosis. This has been confirmed by the availability of practical methods that employ classification, segmentation, or a combination of both to quantitatively and subjectively characterize the brain. The processing of MR images can be performed manually, semi-automatically, or through fully programmed processes based on human interaction. Accurate segmentation and classification are essential for medical image processing, but they must usually be performed manually by specialists, which takes time. Conversely, an accurate diagnosis allows patients to start an appropriate treatment sooner and have longer lifespans. Therefore, in the area of artificial intelligence (AI), there is a pressing need for the development and design of innovative frameworks to decrease the amount of work that radiologists and physicians must perform to diagnose and characterize tumors.

This work uses an automation system that was developed for the purpose of classifying and segmenting brain tumors. This new technology could significantly improve the diagnostic skills of neurosurgeons and other medical professionals, especially with regard to analyzing tumors in ectopic brain regions, which is crucial for early and precise diagnosis. One of the main principles of this research is to promote accessible and efficient communication. To decrease the knowledge gap between medical professionals, the system simplifies the presentation of magnetic resonance imaging (MRI) results. This improved comprehension can enable well-informed decision-making across the board in clinical workflows. Furthermore, the proposed methodology promotes a fully automated approach, minimizing the need for extensive pre-processing steps. This not only optimizes the workflow’s efficiency but also mitigates the potential for human errors during pre-processing, ultimately leading to more consistent and reliable results.

## 4. Proposed Approach

The proposed framework and the experimental stages are described in Figure 1 and discussed in this section.

### 4.1. Proposed Framework

Within the suggested framework, the input images are as shown in Figure 2 and Figure 3, obtained from MRI datasets [29].

First, we apply a median filter to remove noise and then apply augmentation techniques such as brightness and contrast enhancement; we then proceed to the extraction stage by combining EfficientNetV2B3 and the CNN, as well as the KNN classifier, with the exception of the the deep CNN models’ standard softmax classifier. Our studies use KNN because we were inspired by earlier research that showed KNN to perform well in CNN feature classification.

### 4.2. Dataset

We worked with two distinct, publicly accessible brain MRI datasets to perform our experimental studies.

The first dataset consisted of MRI images of three types of brain tumors, namely gliomas, meningiomas, and pituitary tumors. Cheng et al. [29] initially processed the dataset, which was obtained from Tianjin Medical University General Hospital and Nan Fang Hospital in Guangzhou, China. Figure 3 displays a selection of sample images from the dataset, along with their corresponding class labels. The ground truth for the tumor dataset includes the following tumor types: pituitary tumors, gliomas, and meningiomas, with a total of 930, 1426, and 708 images, respectively. The dataset of brain CE-MRI images can be accessed at (https://figshare.com/articles/dataset/brain_tumor_dataset/1512427, accessed on 19 September 2024).

The second dataset, named Brain Tumour Detection 2020 (BR35H) [30], is an MRI dataset obtained from the Kaggle website at (https://www.kaggle.com/datasets/ahmedhamada0/brain-tumor-detection, accessed on 19 September 2024). We refer to this dataset as dataset2, consisting of two classes, with 1500 images of the tumor class and 1500 images of normal or non-tumor cases.

### 4.3. Image Pre-Processing

MRI images contain a significant amount of noise, which could be caused by the surroundings, instrumentation, or operator errors. These could cause significant MRI scan inaccuracies. Thus, the first stage is to clean the MRI image of noise. There are two different types of noise reduction methods: linear and nonlinear. The average weight of the neighborhood is used to update the pixel value in linear filters for noise reduction. The image quality is decreased by this process. Conversely, in the nonlinear method, the delicate structures are degraded but the sides remain intact [31]. The essential challenge in the pre-processing step is to enhance the best MRI brain tumor image and enhance the contrast. The pre-processing step can assist in increasing the signal-to-noise ratio, eliminating noise, and clarifying the edges of the MRI brain tumor image. In this case, the noise was removed from the images using a median filter, as illustrated in Figure 4.

### 4.4. Data Augmentation

Data augmentation is defined as a technique that involves artificially creating updated versions of a dataset to increase the training set and use the available data. It includes making minor modifications to the dataset to generate new data points [32].

Data augmentation has demonstrated strong performance in various clinical image issues and brain scan analysis. The data augmentation method is used to artificially increase the scale of the training image data through rotating and flipping the authentic dataset. More training image data will assist the CNN structure in reinforcing the overall performance and creating skillful models [33]. In this study, the contrast and brightness augmentation methods illustrated in Figure 5 were applied to the images during training.

### 4.5. Feature Extraction

In these experiments, automated feature extraction was achieved by integrating the CNN and EfficientNetV2B3, as seen in Figure 6.

### 4.6. Classification Methods

The algorithm known as KNN is a supervised learning algorithm and a non-parametric method that relies on proximity to perform predictions or classifications of specific data, and it can be used for classification and regression tasks. The principle on which it operates is that similar points are located close to each other. The KNN algorithm has been utilized in the classification of brain tumor datasets. KNN is an appropriate algorithm for multi-class category problems, as KNN exhibits higher accuracy as a classifier. We use the hybrid EfficientNetV2B3 and KNN to enhance the accuracy. The KNN takes the area of the softmax mechanism and the conventional pooling layer within the last image for categorization [34].

The K-nearest neighbors pseudo-code is shown in Algorithm 1.
**Algorithm 1** KNN Algorithm.1. run dist(Z, Zx) *where x = 1,2, …, m;**where dist is the distance between points.*2. *Order the Euclidean distances m.*3. *Let k be +ve INT. Get first k distance from list.*4. *Get k-points corresponding to kdistance.*5. *Let kx be the number of points belonging to**among k, i.e., I ≥ 0* 6. *If kx>ky∀ x ≠ y, then put z in class x.*


### 4.7. Implementation

The image classification performance relies on dividing the image features and performing classification as separate phases within the framework. The classification stage is crucial in brain tumor classification models. The framework’s code is created using Python libraries, and we execute the Python code on Google Colaboratory (Colab). It is tested on two different datasets using various augmentation techniques, such as brightness and contrast enhancement, and without augmenting the input image. The training and execution of the hybrid technique are also performed using Google Colaboratory (Colab). This cloud service is based on Jupyter Notebooks, and it provides a virtual GPU powered by an NVIDIA Tesla K80 with 16 GB RAM. The keras library is adopted, along with TensorFlow, to build the deep learning architecture.

A CNN is a feedforward neural network and is frequently used for the collection of data in order to obtain discriminative information. The convolutional layers are Conv1D (1, 256) and Conv1D (1, 128). The ReLu activation function is used. After processing the data by means of the Conv1D (1, 256), Conv1D (1, 128), and max-pooling layers, the dimensions of the output are 1*128. EfficientNetV2 is considered a new family of convolutional networks that have faster training speeds and better parameter efficiency than previous models; it is used to extract view feature X1. The CNN is applied to extract the shape feature X2. X1 and X2 are processed by the flatten layer.

First, the median filter was used to minimize noise in the input images, and the augmentation methods mentioned previously were applied. Then, the features were extracted using the hybrid method (CNN and EfficientNetV2B3) as a feature extraction layer. The KNN algorithm was executed in the classification stage. The KNN parameters include k, which is equal to 49, and the next neighborhood or distance metrics. A small k value can increase the system’s vulnerability to noise and overfitting. A greater value of k is used to increase the importance of the calculations. Additionally, class-related data imbalances can occur. When k is given a high value, the results become dominant. The experiment was performed five times, and each experiment was performed as follows: five occasions with a validation process. The average result of these five experiments is given as the average value and deviation.

### 4.8. Evaluation Metrics

Precision measures how often the model correctly predicts the disease, and it is calculated using the following equation:$$Precision=\frac{TP}{TP+FP}$$

The following formula can be used to calculate the recall:$$Recall=\frac{TP}{TP+FN}$$
F1 score: Calculated using the following formula, the F1 is the weighted average of the true positive rate (recall rate).
$$F1-score=\frac{2*precision*recall}{precision+recall}$$

## 5. Experimental Results

Multiple metrics have been established for the assessment of a typical classifier’s performance. The most widely used measure of quality is the accuracy of classification. Accuracy measures the percentage of correctly classified samples according to to the total number of samples.

The accuracy of classification achieved on each dataset is shown in Table 1. It is found that the best method, utilizing contrast images on dataset1, achieved accuracy of 99.51%, as shown in Figure 7.

A confusion matrix summarizes the predictions of a model. Each row corresponds to a real category and a single item. A confusion matrix is normalized by dividing each element’s value in every class, enhancing the visual representation of misclassification in each class [35]. Normalized confusion matrices for the best technique are shown, where G, P, and M, or 0, 1, and 2, refer to gliomas, pituitary tumors, and meningiomas, respectively. Various metrics can be used from the confusion matrix to demonstrate the classifier’s performance for each tumor category. Recall (or sensitivity) and precision are essential metrics [36,37,38].

Figure 8 and Figure 9 present the confusion matrices for the proposed framework when using various augmentation techniques on each dataset.

## 6. Ablation Study

To confirm the significance of the proposed approach, ablation experiments were performed with augmentation and the baseline. We evaluated the hybrid classification approach, and the proposed hybrid model’s results are presented in Table 2 and Table 3. The classification experiments were conducted with several configurations, with the hybrid approach implemented in conjunction with extensions and compared to the basic model.

## 7. Discussion

This article introduces a precise and completely automated system, requiring minimal pre-processing, for the categorization of brain tumors. The system utilizes advanced transfer learning techniques to analyze the features of brain MRI scans. KNN is utilized to extract the features. Classifiers rely on augmentation techniques to improve their performance. This framework achieved significantly higher precision when compared to all other relevant options from previous research, and it could be further examined using a more significant number of images.

Typically, the primary metric used to evaluate such systems in the majority of previous studies is the accuracy criterion, as well as using the sensitivity, precision, and recall. We compare the obtained results with those of past work conducted on an identical benchmark dataset [29] in Table 4 and on the Kaggle BR35H dataset in Table 5.

The authors in [10,24,25,26,27,28,39,40,41] achieved accuracy of 91.28%, 84.19%, 91.90%, 93.68%, 90.89%, 97.1%, 97%, 97.25%, and 97.91% respectively. In this study, the suggested model has also achieved high accuracy when tested on the same dataset1, exceeding that of the best-performing model in the previous studies by 1.5%.

The authors in [30,43,44,45] achieved accuracy of 97.5%, 98.8%, 99.50%, and 97.99%, respectively. High accuracy has also been attained with the proposed model when tested on the same BR35H Kaggle dataset2, exceeding that of the best-performing model in the previous studies by 0.3%.

## 8. Conclusions

This paper introduces a system for the automatic categorization of brain tumors, requiring only minimal processing. The training of the model on a brain tumor MRI dataset consisting of 3064 images was assessed with various performance metrics, like the accuracy, precision, and recall. We combined the flatten layer of the CNN with the flatten layer of EfficientNetV2B3 as the feature extraction layers and then used a KNN classifier with various augmentation techniques. This led to 99.51% accuracy, and the use of improved, cutting-edge technology increased this by 1.5%. We intend to expand the proposed framework to more extensive datasets and more brain tumor types. Moreover, the proposed framework will become available in the cloud to provide doctors with fast and accurate diagnoses when using MRI images as input. Future research efforts will extend the proposed framework by incorporating additional types of brain tumors and larger datasets. With the help of the recommended web applications, medical professionals can analyze MRI images with speed and precision. A range of medical imaging techniques, such as X-rays, ultrasound, endoscopic methods, thermoscopy, and histological imaging, can be employed with the proposed model. Future studies could address a number of the constraints of this work. An adequate evaluation requires a large amount of extra patient data, mainly for the meningioma class, which had the lowest number of images of all three training classes studied. Moreover, studies should focus on tuning additional hyperparameters for a variety of convolutional layers, a variety of filters for each convolutional layer, the kernel size, and a variety of absolutely linked layers. In addition, the enhancement of the proposed model may be achieved by increasing the dimensions of the hyperparameters. Finally, by including CE-MRI images of normal brains in the dataset, further differentiation may be achieved for tumor classification. In the future, a larger dataset could be used for educational purposes. Finally, the problems of dimensionality that arise due to moving weights and parameters should be addressed.

## Figures and Tables

**Figure 1 diagnostics-14-02710-f001:**
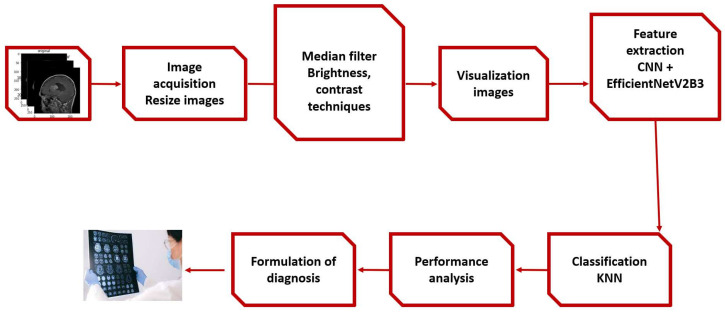
The proposed framework.

**Figure 2 diagnostics-14-02710-f002:**
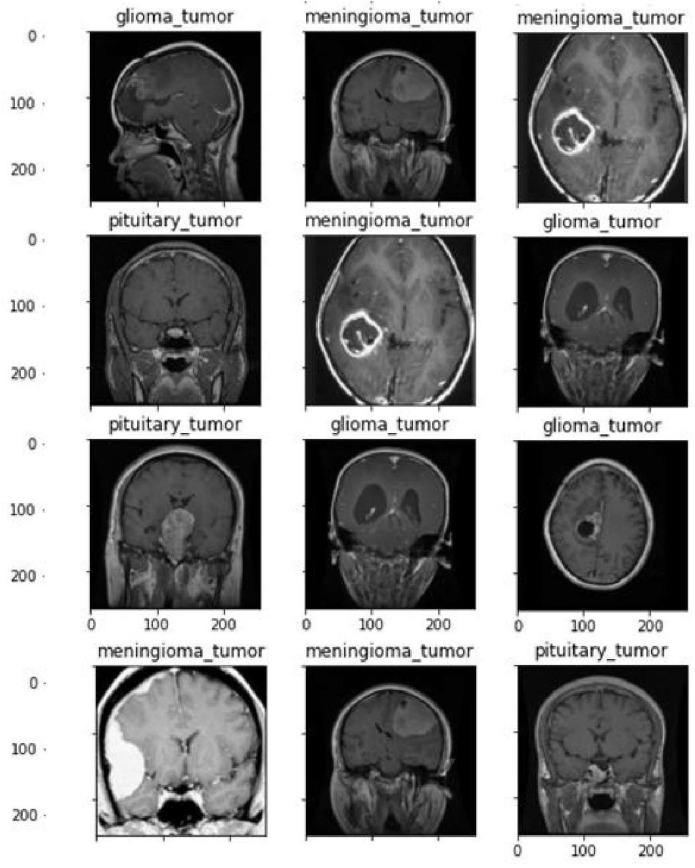
Samples of brain tumor images with class labels in dataset 1.

**Figure 3 diagnostics-14-02710-f003:**
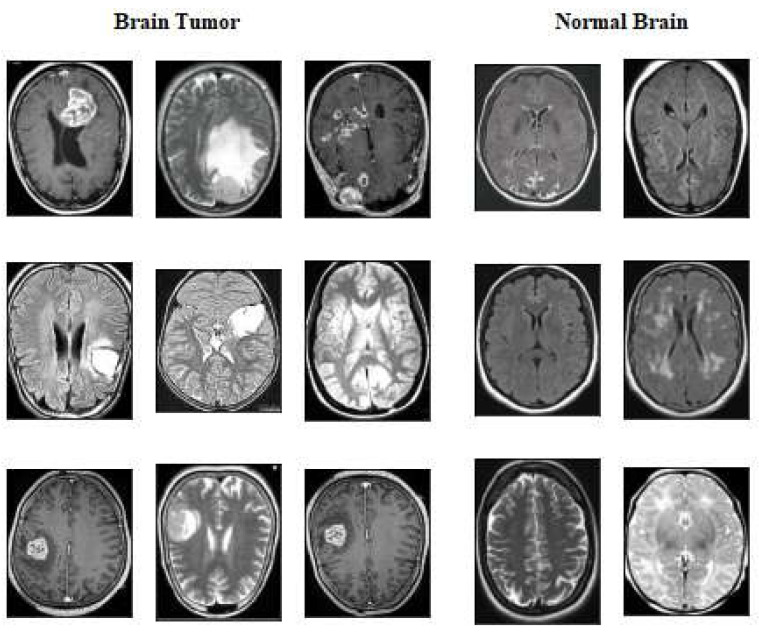
Samples of brain tumor images with class labels in dataset 2.

**Figure 4 diagnostics-14-02710-f004:**
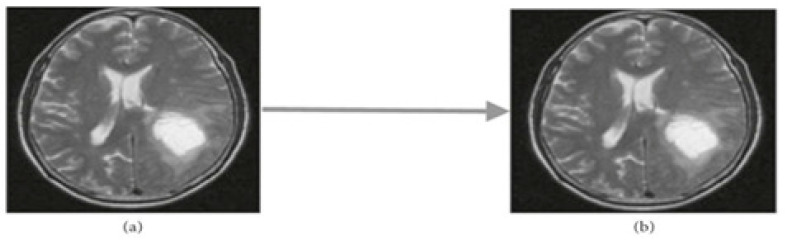
Applying median filter. (**a**) Input image. (**b**) Median Filtering.

**Figure 5 diagnostics-14-02710-f005:**
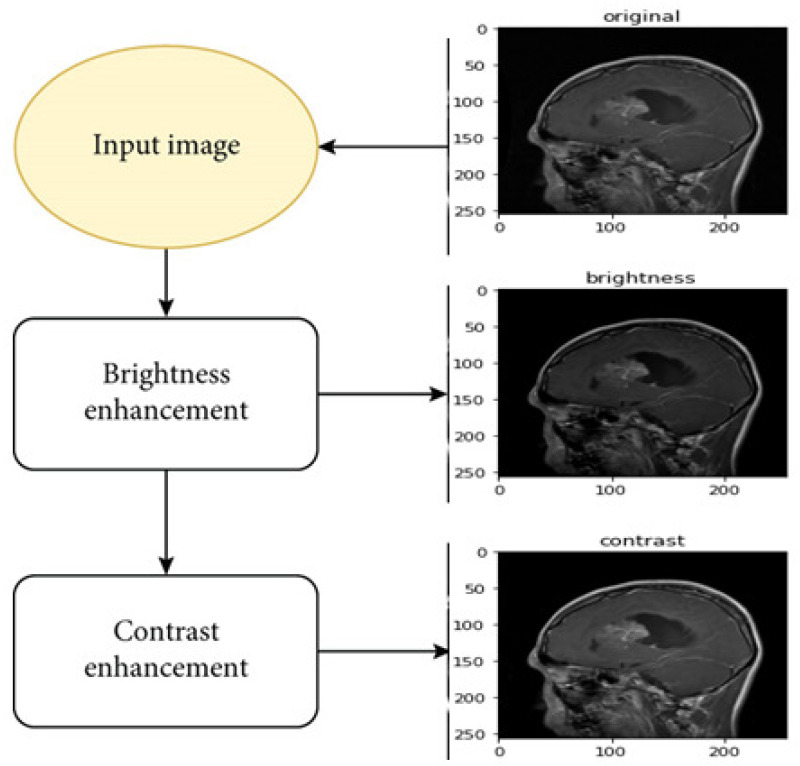
Applying augmentation techniques including brightness and and contrast enhancement.

**Figure 6 diagnostics-14-02710-f006:**
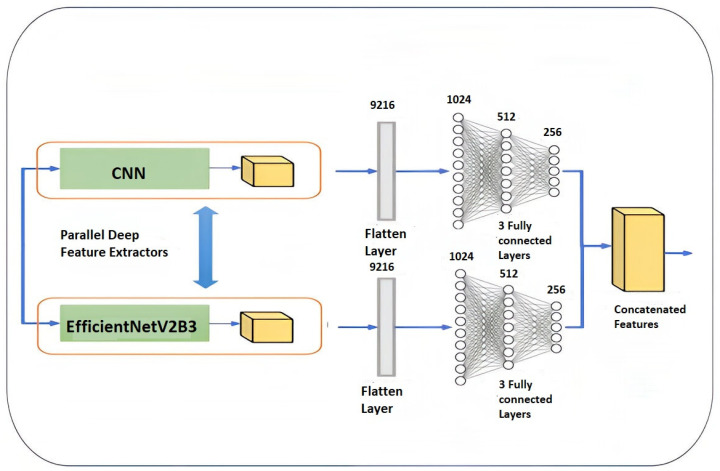
Feature extraction layer followed by KNN classifier.

**Figure 7 diagnostics-14-02710-f007:**
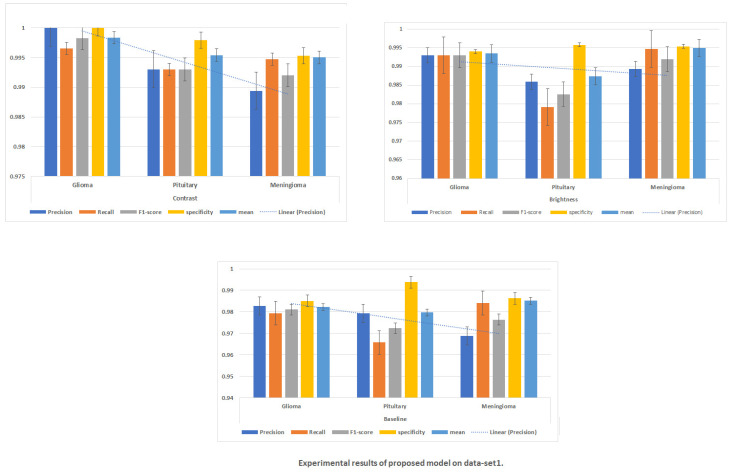
Experimental results of proposed model on dataset1.

**Figure 8 diagnostics-14-02710-f008:**
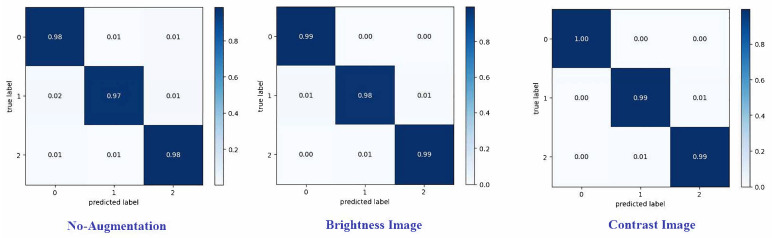
Confusion matrix for proposed model on dataset1 with different augmentation techniques.

**Figure 9 diagnostics-14-02710-f009:**
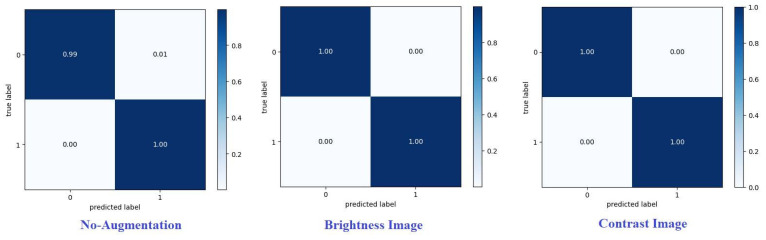
Confusion matrix for proposed model on dataset2 with different augmentation techniques.

**Table 1 diagnostics-14-02710-t001:** Experimental results of proposed model.

Dataset	Augmentation	Precision	Recall	F1 Score	Specificity	Mean
Dataset1	Baseline	99.76%	97.64%	97.66%	98.83 %	98.23%
	Brightness	99.02%	98.89%	98.92%	99.50%	99.19%
	Contrast	99.51%	99.47%	99.44%	99.77%	99.19%
Dataset2	Baseline	99.46%	99.47%	99.46%	99.46 %	99.47%
	Brightness	99.83%	99.83%	99.83%	99.83%	99.83%
	Contrast	99.83%	99.83%	99.83%	99.83%	99.83%

**Table 2 diagnostics-14-02710-t002:** Results of experiments on dataset 1.

Augmentation	Tumor	Precision	Recall	F1 Score	Specificity	Mean
Contrast	G	1	0.9965	0.9982	1	0.9983
	p	0.993	0.9930	0.9930	0.9979	0.9954
	M	0.9894	0.9947	0.9920	0.9953	0.9950
Brightness	G	0.9930	0.9930	0.9930	0.9940	0.9935
	p	0.9860	0.9792	0.9826	0.9958	0.9874
	M	0.9894	0.9947	0.9920	0.9954	0.9950
Baseline	G	0.9828	0.9794	0.9811	0.9851	0.9822
	p	0.9792	0.9658	0.9724	0.9938	0.9797
	M	0.9688	0.9841	0.9764	0.9863	0.9852

**Table 3 diagnostics-14-02710-t003:** Results of experiments on dataset 2.

Augmentation	Tumor	Precision	Recall	F1 Score	Specificity	Mean
Contrast	Tumor	1	0.9983	1	1	0.9983
	No tumor	0.9967	1	0.9983	0.9967	0.9983
Brightness	Tumor	0.9993	0.9973	0.9983	0.9993	0.9983
	No tumor	0.9973	0.9993	0.9983	0.9993	0.9983
Baseline	Tumor	0.9973	0.9921	0.9947	0.9973	0.0.9947
	No tumor	0.9921	0.9973	0.9947	0.9921	0.9947

**Table 4 diagnostics-14-02710-t004:** Comparison of the obtained accuracy with that achieved in past work on the same dataset (dataset1).

Author	Year	Method	Performance
Cheng et al. [24]	2015	BoW-SVM	91.28%
Paul et al. [39]	2016	Fully connected CNN	84.19%
Ismael et al. [25]	2018	DWT-Gabor-NN	91.9%
Pashaei et al. [26]	2018	CNN-ELM	93.68%
Afshar et al. [40]	2019	CapsNet	90.89%
Deepak et al. [41]	2019	Deep CNN-SVM	97.1%
Ismael et al. [10]	2020	ResNet50	97% for image level
Gull et al. [27]	2021	VGG-19, AlexNet	97.25%
Mondal et al. [28]	2022	DenseNet201, InceptionV3, MobileNetV2, ResNet50, and VGG19	97.91 %
Eman et al. [42]	2023	ResNet50+KNN	99.1%
Proposed model	2024	Combination of CNN with EfficientNetV2B3 and KNN classifier	99.52%

**Table 5 diagnostics-14-02710-t005:** Comparison of the obtained accuracy with that achieved in past work on the same dataset (dataset2 (BR35H)).

Author	Year	Method	Performance
Hamada et al. [30]	2020	CNN models	97.5%
Asmaa et al. [43]	2021	CNN with augmented images	98.8%
Amran et al. [44]	2022	AlexNet, MobileNetV2	99.51%
Falak et al. [45]	2023	Keras Sequential Model (KSM)	97.99%
Proposed model	2024	Combination of CNN with EfficientNetV2B3 and KNN classifier	99.83%

## Data Availability

The data collected and analyzed during this study are available from the corresponding author upon reasonable request.

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
