# Peer review of "A Hybrid Deep Learning Model with Data Augmentation to Improve Tumor Classification Using MRI Images"

_diagnostics, 2024, doi:10.3390/diagnostics14232710_

Round 1
Reviewer 1 Report
Comments and Suggestions for Authors
In this study, the authors have proposed a hybrid deep learning model to improve tumor classification using MRI images in order to address the knowledge gaps in detecting brain cancer. They used transfer learning using the weights of EFFICIENTNET V2B3 model for feature extraction. For classification, they used K-neighbor network algorithm. The model was validated using two datasets.
The study does not provide a novel method for tumor classification. However, they proposed a customized architecture for feature extraction. It can be considered as a novel technique that addresses the gap in existing literature.
Compared to existing literature, the study proposed an effective feature extraction (as per the study outcomes) that improves the tumor classification using the MRI images.
The results were impressive. However, the authors have to provide the computation costs for tumor classification. They showed that there is a significant improvement in the proposed model by achieving an accuracy more than 99% which is impossible without huge computational costs. The KNN model is not efficient than gradient boosting based classification. In addition, CNN is an efficient network that extracts features without any dedicated preprocessing tasks. Why the authors add an additional layer of pre-processing in the proposed study. It adds a substantial computational overhead to the proposed model.
They validated the proposed model using two datasets. They obtained an exceptional accuracy of 99.52 % and 99.83 % on datasets 1 and 2, respectively. The conclusions are consistent. However, additional experiments are required to show the model is efficient in resource-constrained environment. The proposed model can support clinicians in making effective decisions. However, without any model interpretability, it is impossible to implement in healthcare settings. Moreover, the authors does not provide detailed information regarding feature concatenation. During the concatenation, the less informative features can be added. It can influence the model performance. How the authors have addressed the limitations.
All the references are appropriate.
Tables are placed correctly. However, figure no. 4 is not necessary. The quality of figures no. 6-9 can be improved.
Please correct English.
Author Response
Please find attached response file

Reviewer 2 Report
Comments and Suggestions for Authors
In this article, the authors suggested a hybrid deep learning model for tumour classification. The paper requires major revision in order to be published in this journal.
1. In Abstract, there is definition for KNN.
2. In introduction, the authors need to explain the study's objective in detail.
3. The research methodology demands a major revision in order to present the novelty of this research. As deep learning does not require a dedicated data preprocess, why the authors performed multiple data-preprocess. How they overcome the challenges in integrating EfficientNet V2 B3 and KNN.
4. EfficientNet models are computationally intensive. There is no details associated with computational costs.
5. The authors need to provide a dedicated discussion section, discussing the study implications, limitations and future direction.
Comments on the Quality of English LanguageModerate English Revision is required.
Author Response
Please find the attached response file

Round 2
Reviewer 2 Report
Comments and Suggestions for Authors
The authors have addressed my concerns. The manuscript may be acceptable in the current form.